# A Case Report of Posterior Reversible Encephalopathy Syndrome (PRES) in a Nonsevere Case of COVID-19

**DOI:** 10.3390/brainsci12070915

**Published:** 2022-07-13

**Authors:** Małgorzata Cisowska-Adamiak, Katarzyna Sakwińska, Iwona Szymkuć-Bukowska, Anna Goclik, Iwona Lunitz, Magdalena Mackiewicz-Milewska

**Affiliations:** Department of Rehabilitation, Nicolaus Copernicus University in Toruń, Collegium Medicum in Bydgoszcz, 85-094 Bydgoszcz, Poland; k.sakwinska@cm.umk (K.S.); iwona.szymkuc@cm.umk.pl (I.S.-B.); agoclik@cm.umk (A.G.); iwona.lunitz@cm.umk.pl (I.L.); magmami@onet.eu (M.M.-M.)

**Keywords:** posterior reversible encephalopathy syndrome, PRES, COVID-19, case report

## Abstract

Posterior reversible encephalopathy syndrome (PRES) is a rare complication that the exact pathophysiological mechanism of which is still unclear. PRES most often occurs in connection with severe hypertension and autoimmune diseases. It can also appear during chemotherapy or immunosuppressive treatment. A 38-year-old woman with a negative medical history was admitted to the local hospital due to loss of consciousness accompanied by seizures and high values of blood pressure, and a PCR test for COVID-19 was positive. The patient’s condition was preceded by weakness, wet cough, runny nose, and low-grade fever for three days. Due to the conducted diagnostics after negative CT scans and angio CT studies, an MRI of the head with contrast was performed, where changes characteristic of PRES syndrome were found. During the hospitalization, the patient did not require invasive ventilation and did not receive antiviral drugs or tocilizumab as a result of treatment for her high blood pressure values, and after establishing the diagnosis, the patient was discharged home with a significant improvement in her well-being. In the literature, there are discussions as to whether COVID-19 predisposes patients to PRES. Isolated cases have been described, but its frequency is not yet established. Case reports in the literature appear to be specifically associated with a severe course of the disease, unlike in our patient. Even with a mild course of COVID, the diagnosis of PRES should be taken into account in patients with seizures, visual disturbances, or other focal neurological deficits.

## 1. Introduction

Posterior reversible encephalopathy syndrome (PRES) can manifest as one or more signs, such as headaches, focal neurological deficits, seizures, visual disturbances, encephalopathy, and other focal neurological deficits [1,2]. Its etiology is not fully understood, and one of the hypotheses is that in this syndrome, there is a disturbance in vascular autoregulation as a result of a sudden increase in blood pressure, while another suggests that epithelium dysfunction occurs in response to severe infections or toxins [2]. PRES most often occurs in connection with severe hypertension and autoimmune diseases. It can also appear during chemotherapy or immunosuppressive treatment [3]. Magnetic resonance imaging (MRI) demonstrates bilateral cortico-subcortical T2 confluent hyperintensities in the parietal and occipital lobes and less frequently in the frontal lobes, temporal lobes, or cerebellum. The characteristic feature of this syndrome is that after beginning treatment for the probable cause, the neuroimaging changes visible in the CNS usually completely disappear [3].

To date, a few individual cases and series of cases have been described in the literature in which PRES was diagnosed during the course of COVID-19 infection, but those cases were related to a very severe course of disease that required intensive treatment, including intubation. By contrast, our patient maintained respiratory efficiency throughout their entire stay in the department and had no other characteristic signs of COVID-19.

There are ongoing discussions as to whether COVID-19 can be considered to increase predisposition to this syndrome more than other severe diseases. Thus, it can be speculated that the causes of its occurrence are disorders resulting from COVID-19, such as disordered cerebrovascular autoregulation, acute renal failure, acute hypertension, hypoxia, inflammation, and endothelial injury [2].

## 2. Case Report

A 38-year-old woman with a negative medical history and vaccinated twice: December 2021 and January 2022, was brought to the local hospital by an ambulance service due to loss of consciousness and seizures while in a lying position in bed without biting her tongue or involuntary urination.

Three days before the incident, the patient developed weakness, wet cough, runny nose, and low-grade fever. On the day of admission at about 5 a.m., she suddenly fell down on the way back from the toilet to bed. There were convulsions during the incident, as reported by the parents, and the patient was unconscious. After the arrival of the emergency medical team, she refused to be transferred to the hospital. An antigen test for Sars-Cov-2 was performed, which was positive. Around noon, another loss of consciousness with seizures in the patient occurred in the presence of her parents and did not include head trauma. After the event, the woman developed amnesia and confusion. An ambulance was called again, and the patient was transported to the hospital. The paramedics did not administer any anti-epileptic drugs because the seizures ended before their arrival.

On admission to the hospital, a PCR test was performed, which confirmed the earlier positive COVID-19 result. Due to the earlier loss of consciousness and convulsions, a CT scan of the head was performed, which showed no abnormalities. Inflammatory foci found via High-Resolution Computed Tomography (HRCT) of the lung characteristic for COVID-19 represented less than 5% of lung volume. After the admission, the patient was conscious and did not demonstrate any neurological symptoms. She denied dyspnea and did not require oxygen therapy. In the first hours of hospitalization, the patient reported epigastric pain in the form of stinging in addition to nausea. An ultrasound and X-ray of the abdomen was performed at that time to exclude obstruction, perforation, or abscess and did not show any deviations. Elevated blood pressure values were observed, and antihypertensive treatment was initiated, but despite this, blood pressure values continued to increase in the following hours.

The next morning after admission, two seizures were again observed with subsequent cognitive impairment and somnolence for several hours. Levetiracetam was included in the treatment. Three days after admission, the patient reported sudden unilateral vision loss. Ophthalmological examination showed no abnormalities. This symptom disappeared the following day.

Computed tomography (CT) angiography of the head and CT with contrast were performed and excluded aneurysm, arteriovenous malformation (AVM), dural sinus thrombosis, and tumor. Due to the lack of deviations in the previous tests, an MRI of the head with contrast was performed, in which small areas and hyperintense bands in the T2-weighted image (T2-WI) and Fluid Attenuated Inversion Recovery (FLAIR) were visualized on both sides in the parieto-occipital areas without either diffusion restrictions or contrast enhancement, a typical manifestation of PRES [4], as shown in Figure 1, Figure 2, Figure 3 and Figure 4.

In order to determine the cause of the patient’s high blood pressure, an ultrasound of the abdomen and Doppler of the renal arteries, an abdominal CT scan with the adrenal glands in search of focal lesions, and CT angiography of the renal arteries were also performed. Neither ultrasound nor CT angiography showed any signs of stenosis of the renal arteries. In the following days of hospitalization, blood pressure decreased, and a reduction in the dose of antihypertensive treatment was thus necessary (Table 1).

Throughout the patient’s stay in the Isolation Department, the maximum CRP (C Reactive Protein) value was 27.56 mg/L (normal up to 5 mg/L), and the value on the day of admission was 3.15 mg/L. There was a transient increase in AST (Aspartate Transaminase) (max 564; normal up to 34 U/L) and ALT (Alanine Aminotransferase) (max 189; normal 10–40 U/L). High CPK (creatine kinase) values up to 26123 U/L (normal 29.0–168.0 U/L) were observed. The maximum troponin levels were 132 ng/L (normal < 38.6). All of these laboratory parameters were normalized prior to discharge. Negative procalcitonin and normal TSH, glucose, and creatinine levels were recorded throughout the hospitalization period. The examination of the cerebrospinal fluid (CSF) did not reveal deviations, as shown in Table 1.

Due to the unclear causes of hypertension, basic hormonal profiling was performed, showing abnormal cortisol values in the daily rhythm (38.3 µg/dL, normal 5.3–22.5) and abnormal cortisol values in the dexamethasone inhibition test (20 µg/dL). Additionally, a slight increase in cancer antigens was observed. CEA (Carcinoembryonic Antigen) 16.9 (normal 0–2.5 ng/mL), CA19-9 (Carbohydrate antigen 19-9) (CA19-9) 35.2 U/mL (normal < 30.9 U/mL), AFP (Alpha-fetoprotein), and B-HCG (Human Chorionic Gonadotropin) were normal. Further diagnosis was recommended. Diagnostics were performed as in the attached Table 2.

Throughout the entire hospitalization, the patient did not require oxygen therapy and was not treated with antiviral drugs. No seizures were observed after the initiation of anticonvulsant therapy. She was sent home in good general condition with recommendations for further digestive and endocrine diagnostics. There were no adverse or unanticipated no events throughout the stay. The final diagnosis at discharge was PRES, the prognosis of which was good.

In a telephone conversation with the patient conducted two and a half months after discharge from the hospital, the patient reported improved wellbeing. Her arterial pressure is regularly monitored, and she required further reductions in the prescribed antihypertensive drugs under the supervision of her GP (general practitioner). She has a scheduled MRI for her head in the near future and is in the process of undergoing the recommended diagnostics. 

## 3. Discussion

A review of brain imaging studies in 2054 patients with COVID-19 found that head imaging was required in 1.1% of cases (278 patients) during the course of infection [5]. The patient we reported is the only one in whom we diagnosed PRES among 180 patients hospitalized in the COVID-19 Department, which was set up from mid-November 2021 to the end of March 2022, i.e., during the COVID-19 Delta and Omicron variants waves. The patient developed symptoms characteristic of PRES, namely seizures, which occur in 80% of PRES patients [6], and visual disturbances (39%) [7].

In the literature that is available to date, patients with PRES syndrome have so far been reported to have very severe COVID-19 infection that requires treatment in the ICU including intubation, therapy with multiple anti-COVID-19 drugs, and broad-spectrum antibiotic therapy [8,9,10]. In some patients, the observation of symptoms was challenging due to their severe general condition affection.

It is unusual that our patient, unlike other cases in the literature, was not critically ill from COVID-19, did not receive antiviral drugs or tocilizumab, and did not require invasive ventilation. 

The limitation of this study was the lack of analysis of the possible autoimmune causes and the fact that we do not have any MRIs of control heads in which there should be no more focal changes. 

In our patient, the development of PRES syndrome could be linked with her high blood pressure values, an association that is confirmed in the literature [11]. However, from our point of view, the question remains as to whether COVID-19 was the trigger for the rise in blood pressure in this case, or whether it was rather that symptoms were a direct effect of viremia on the endothelium.

A possible response is the mechanism of SARS-CoV-2 binding to angiotensin-converting enzyme 2 receptors, which can dysregulate the endothelial layer, increase blood pressure, and disrupt cerebral blood flow autoregulation [12]. 

The issue of abnormal cortisol levels and the outcome of dexamethasone inhibition regarding whether the transient hormonal disruptions were the result of COVID-19 infection or not remains unresolved. The patient did not morphologically appear to have Cushing’s syndrome: slim, without diabetes, and no hypertension prior to infection. There are no similar studies in the literature on disorders of cortisol secretion during the course of this infection.

We want to point out in our case report, that even in with a mild course of COVID, the diagnosis of PRES should be taken into account in the patients with symptoms such as seizures, visual disturbances, or other focal neurological deficits. Considering this diagnosis and conducting appropriate tests in this direction is important for the proper treatment and monitoring of the patient. Due to the correct diagnosis and treatment, the patient experienced a significant improvement in a short time. After being discharged from the hospital, she was able to return to her life and work.

## 4. Conclusions

COVID-19 infection, even when its course is not severe in terms of lung involvement, can cause other dangerous complications. One such example is PRES, which, as we found out, can occur in nonsevere cases of COVID-19 infection. Awareness of this is important for the timely diagnosis and treatment of this reversible neurological syndrome.

## Figures and Tables

**Figure 1 brainsci-12-00915-f001:**
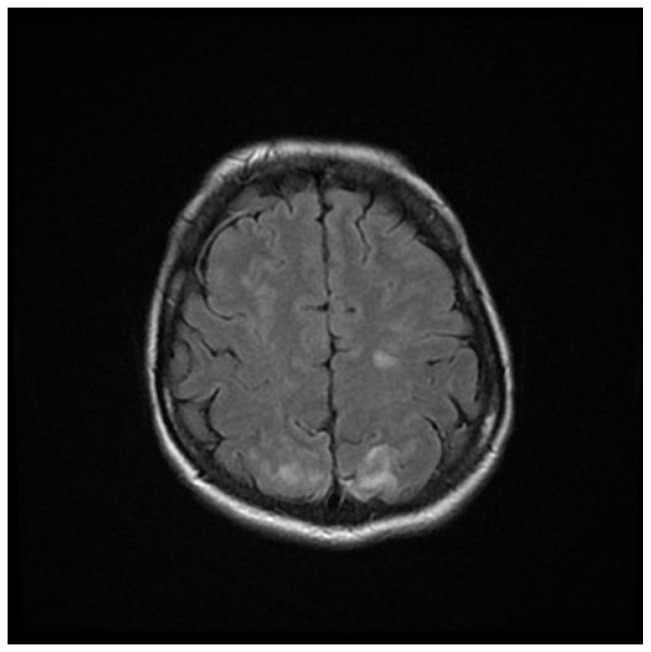
FLAIR MRI sequence of the brain showing hyperintense bands in the parieto-occipital areas on both sides.

**Figure 2 brainsci-12-00915-f002:**
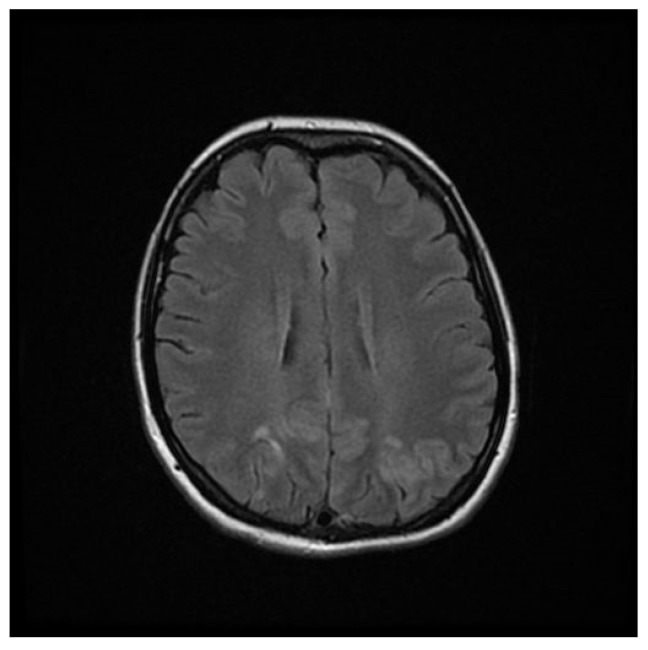
FLAIR MRI sequence of the brain.

**Figure 3 brainsci-12-00915-f003:**
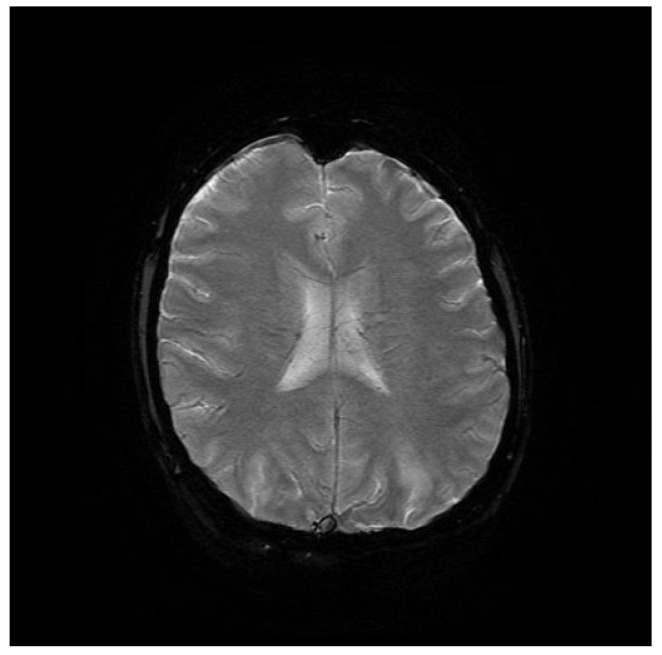
T2-WI MRI sequence of the brain in transverse plane.

**Figure 4 brainsci-12-00915-f004:**
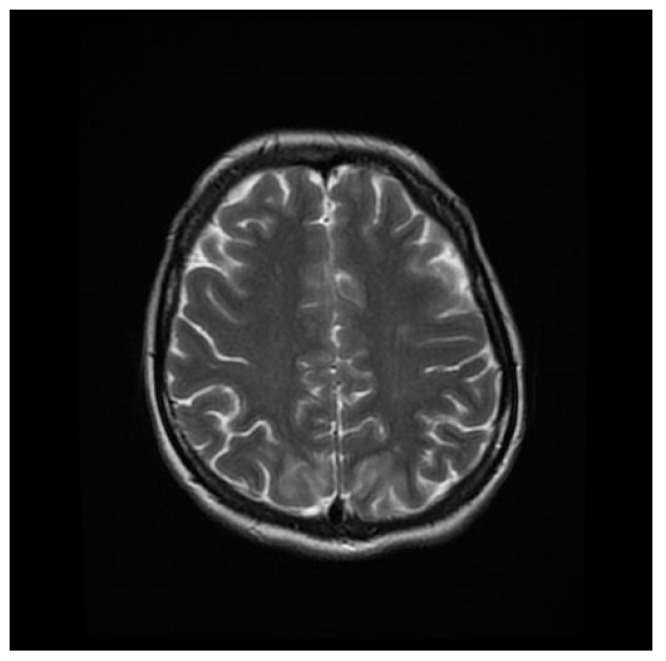
T2-WI MRI sequence of the brain in horizontal plane.

**Table 1 brainsci-12-00915-t001:** CSF analysis.

CSF Analysis	Range	Normal Range
Color	clear	clear
RBCs count	nil *	nil *
WBC count	3	0–5
CSF Proteins	37.7 mg/dL	15–40 mg/dL
CSF glucose	73.6 mg/dL	50–80 mg/dL
Microbial examination	no microorganism	no microorganism

* zero (0).

**Table 2 brainsci-12-00915-t002:** Performed diagnostics.

Type of Examination	Differential Diagnosis	Abnormalities
**Diagnostics the causes of seizures**		
CT scan of the head	Hemorrhagic changes (stroke, SAH)	No
CT angiography of the head	Aneurysm, AVM, dural sinus thrombosis	No
CT of the head with contrast	Tumor	No
MRI of the head with contrast	Tumor, dural sinus thrombosis, ischemic changes, venous stroke, other focal changes, brain inflammation	Small areas and hyperintense bands in T2-weighted image (T2-WI) and FLAIR on both sides in the parieto-occipital areas
The examination of the cerebrospinal fluid	Neuroinfection	No
**Diagnostics the causes of high blood pressure**		
Doppler of the renal arteries	Stenosis of the renal arteries	No
Abdominal CT scan with contrast	Focal lesions of adrenal glands	No
CT angiography of the renal arteries	Stenosis of the renal arteries.	No
Lung HRCT	Severe COVID inflammation	No
Abdomen ultrasound	abscess	No
Abdomen X-ray	Obstruction, perforation	No
Basic hormonal profiling	Cortisol values disorders	Abnormal cortisol values; abnormal cortisol values in the dexamethasone inhibition test
Neo markers	Paraneoplastic syndrome	Slight increase

SAH—subarachnoid hemorrhage, HRCT—High-resolution computed tomography.

## Data Availability

The data presented in this study are available on request from the corresponding author.

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
