# Peer review of "A Case Report of Posterior Reversible Encephalopathy Syndrome (PRES) in a Nonsevere Case of COVID-19"

_brainsci, 2022, doi:10.3390/brainsci12070915_

Round 1

Reviewer 1 Report

I read the interesting case report about Posterior reversible encephalopathy syndrome 3 (PRES) in a nonsevere case of COVID-19.

Here are my comments: 

1- Line Nr. 17: please add the word ''Blood'' to ''high pressure''.

2- In the case presentatoin in the abstract, the authors didnot mention that the patient had COVID-19 infection. They started in Line Nr. 20 mentioning COVID 19. 

3- In Line Nr. 35: pLease replace 'and' with 'or'

4- In Line Nr. 49: please delete the word 'so'

5- Line 56: do the authors mean post ictal state intead of Twilight state?

6- Line 60: please use 5 am instead of 5:00

7- I recommend to reorganise the first paragraph of the case report chronologically, otherwise the readers will lose the order of the occasions. 

8- Line 65-66: what is mean by earlier in '' which confirmed the earlier positive COVID-19 result.''? The authors may write the PCR Test showed positive RNA or were positiv. 

9- Line number 68: what is mean by  HRCT? please address

10- Line 72 : Why ''The ultrasound and X-ray of the abdomen '' were performed? Is this related to the case? or as a routine examination,

11- Table is not actually needed or significant to the readers.

12- Line 66: please add native to the word CT

13- Line 87: please refere to the figure in text like; as shown in Figure (1)

14- Line 87: the sentence: ''MRI was successful, although access for COVID patients was limited due to the lengthy disinfection process'' is not significant to the readers.

15- Line 90: Please add the word ''of the brain' after the word 'MRI sequence'

16- Line 85: please add the meaning of FLAIR and T2-WI, what they stands for. 

17- Line 94 please add the word 'also' after the word 'were'

18- Line 95: the sentencce' . Pyloric 96 wall thickening up to 13mm was found (further diagnostics indicated).'' are non significant.

19- In Line 116: please delete the word ''no'

20- Line 128: please add the word 'COVID-19' before 'Delta' and the word 'variants' after Omicron

21- Line 129: Please add the word 'PRES' after ''80% of'

22- Line 131: please use the word 'date' instead 'us', and word 'patients' instead the word 'people'.

23- Line 135: please add the word 'affection after the word ' condition'.

24: Line 136: please use 'unusual' instead of 'seems important''

 25- Line 138: the sentence '' Before the present hospitalization, she had not been treated for any other diseases and had never been diagnosed with hypertension.'' is mentioned before, you can delete it.

26- Line 150 & 151: I would recommend to reform the text as follows: ''The issue of abnormal cortisol levels and the outcome of dexamethasone inhibition are still unresolved if the transient hormonal disruptions were as a result of COVID-19 infection or not.''

Reviewer 2 Report

1. The permission from the patient should be obtained and uploaded as non-published material. It is advised to have good-quality photos of this document.

2. It is advised to modify the title to ‘‘A Case Report’’ in the end. This change is due to searching google strategies.

3. The manuscript needs to be thoroughly revised some structures are not common to scientific articles.

E.g.

line 11 – ‘‘etiology [1], [2].’’

Line 87 – ‘‘characteristic of PRES syndrome [4]’’

Simera I, Moher D, Hoey J, Schulz KF, Altman DG. The EQUATOR Network and reporting guidelines: Helping to achieve high standards in reporting health research studies. Maturitas. 2009 May 20;63(1):4-6. DOI: 10.1016/j.maturitas.2009.03.011. Epub 2009 Apr 15. PMID: 19372017.

4. Abstract. The abstract should include all important and specific information about the subject being reported.

The following structure should be removed: ‘‘This Case-report was 25 developed using the Care Checklist.’’

5. Neuroimage. The authors should upload more figures in different sequences and planes. The present image is insufficient.

6. The subject's vaccination history or dates of the occurrence of the event should be written.

7. The investigation of other causes should be provided in a table with ranges.

Round 2

Reviewer 2 Report

The Reviewer would like to congratulate the authors for their hard work in the improvement of the manuscript.

1. Some grammatical reviews are needed throughout the manuscript.

L21 ‘‘or Tocilizumab As a result’’ – capitalization

L82 ‘‘following hours .’’ – space

L152 ‘‘increase in neo markers was observed’’ – jargon

2. The authors should revise the following sentence ‘‘changes visible in the CNS usually’’. What are the changes? Neuroimaging, electrodiagnostic, ….

3. What were the medications in use by the individual when he had the seizure? Could these medications not be related to PRES?

4. The description of CSF analysis as well as blood glucose should be included in the manuscript.

5. It is advised to provide the brain MRI in one figure with multiple images. The authors should provide diffusion-weighted imaging to confirm the diagnosis of PRES.

https://www.google.com/search?q=figure+multiple+image
6. How were possible autoimmune causes of PRES assessed?

7. Could the authors provide a table with other cases already reported in the literature?
